# Learning effect on an isokinetic knee strength test protocol among male adolescent athletes

**Daniela C. Costa**[1,2,3]☯, **João Valente-dos-Santos**[2,4], **Jorge M. Celis-Moreno**[1],
**Paulo Sousa-e-Silva**[1,2], **Diogo V. Martinho**[1,2], **João P. Duarte**[1,2], **Tomás Oliveira**[2], **Oscar M. Tavares**[5], **Joaquim M. Castanheira**[2,5], **Rui Soles-Gonçalves**[5], **Telmo Pereira**[2,5],
**Jorge Conde**[5], **Daniel Courteix**[3], **David Thivel**[3], **Manuel J. Coelho-e-Silva**[1,2]☯ *

**1** University of Coimbra, FCDEF, Coimbra, Portugal, **2** University of Coimbra, CIDAF (uid/04213/2020), Coimbra, Portugal, **3** Laboratoire des Adaptations Métaboliques à l'Exercice en Conditions Physiologiques et Pathologiques (AME2P), Université Clermont Auvergne, Clermont-Ferrand, France, **4** Lusophone University of Humanities and Technologies, CIDEFES, Lisboa, Portugal, **5** Polytechnic Institute of Coimbra, Coimbra Health School, Coimbra, Portugal

☯ These authors contributed equally to this work.
* mjcesilva@hotmail.com

**Data Availability Statement:** All relevant data are within the paper and its Supporting Information files.

## Abstract

Learning effect occurs when the best performance is not achieved at the earliest trial of a repeated protocol of evaluation. The present study examined, within testing session, the intra-individual variation in an isokinetic strength protocol composed of five reciprocal concentric and eccentric contractions of knee extensors (KE) and knee flexors (KF) among male adolescent swimmers. Additionally, test-retest reliability was determined as intra-individual mean differences between two consecutive testing sessions. The sample included 38 swimmers aged 10.1–13.3 years. A subsample (n = 17) completed a second visit. Isokinetic dynamometry was used to assess concentric and eccentric contractions of KE and KF at an angular velocity of 60°.s⁻¹. The protocol included three preliminary repetitions that were not retained for analysis, a 60-second interval, and five reciprocal maximal concentric contractions (cc). The preceding sequence was repeated for eccentric contractions (ecc) of KE and KF. Multilevel regression confirmed intra-individual and inter-individual levels as significant sources of variance in peak torque (PT) values. Intra-class correlation (ICC) fluctuated between 0.582 and 0.834 and, in general, a substantial percentage of participants need more than three repetitions to attain their best PT: KEecc (36.8%), KEecc (23.7%), KFcc (39.5%), KFecc (18.4%). For the subsample of 17 swimmers who completed a second testing session, intra-individual mean differences of the best PT were trivial or small. In summary, the validity of shorter protocols may be compromised if swimmers do not attain their best peak torque in the first few attempts, and the reliability of a 5-repetition protocol seemed acceptable.

## Introduction

Muscular strength is defined as the ability to exert force against an external resistance and refers to an essential component of physical fitness [1]. Static strength corresponds to the force

**Funding:** The authors received no specific funding for this work.

**Competing interests:** The authors have declared that no competing interests exist.

exerted against an external resistance without changes in muscle length, while dynamic strength implies modifications of the length. Meanwhile, in human movement, the force is applied distal to the axis of rotation and, consequently, the assessment of angular movement requires standardized ergometers to assess torque values [2]. Torque refers to the product of force and the perpendicular distance from the line of action of the force to the axis of rotation. Isokinetic dynamometry measures the strength of a muscle group through a range of motion at a constant speed. This type of testing is often used in sports medicine and rehabilitation settings to assess muscle function and identify areas of weakness [3]. Peak torque (PT) is probably the most used indicator from isokinetic dynamometry and refers to the maximum torque value obtained from an isokinetic curve [4]. In order to obtain an objective and accurate measurement of peak torque, the isokinetic strength protocol must be valid and reliable.

The number of repetitions required to obtain the peak torque is not consistent in the literature. For example, in the Ghent Youth Soccer Project [5], the authors assessed the isokinetic strength of knee muscle actions using five continuous maximal concentric reciprocal contractions of the knee extensors (KE) and flexors (KF) in a sample of soccer players aged 11.0–13.9 years. Another study [6] tested the isokinetic strength of middle-school boys based on three continuous maximal trials. Importantly, a learning effect is present when the best performance is registered after the earliest repetitions [7]. Then, if the testing protocol assumes an economic format with only three repetitions, the data would probably not reflect the maximal individual capacity [7]. Briefly, the learning effect refers to an improvement in performance that occurs due to practice or experience, rather than due to an actual increase in strength. This means that with repeated trials, participants may become more familiar with the task and more efficient in their movements, resulting in increased strength values [8].

A recent study modelled longitudinal changes in knee muscle isokinetic strength in young soccer players and concluded that, on average, annual gains were 5.4 N.m and 2.7 N.m for KE and KF, respectively. To ensure data quality in isokinetic strength, it is important to adopt standardized protocols [5]. A previous study [9] examined day-to-day variation of isokinetic knee concentric and eccentric muscular actions among adolescent basketball players aged 14–16, with intra-class correlations (ICC) fluctuating from 0.72 to 0.89. The ICC is often used to report data quality from repeated testing sessions. In another study, ICC values for KEcc and KFecc ranged from 0.78–0.89 among youth female soccer players [10]. This learning effect in isokinetic testing of knee muscle actions in young athletes has not been systematically addressed within a testing session.

In that context, the present study aimed to determine the learning effect of an isokinetic strength protocol composed of five reciprocal repetitions of knee muscle actions in male adolescent swimmers aged 10.1–13.3 years. Additionally, the test-retest reliability was calculated for the best PT value on two independent testing sessions. It was hypothesized that a 5-repetition protocol is an acceptable methodology to mitigate the learning effect and ensure data quality. The learning effect and data quality of the best PT are expected to fluctuate according to the solicited muscle group and type of contraction.

## Materials and methods

### Procedures

The present study has been conducted in accordance with the ethical standards established for human studies [11], and was approved by the Ethics Committee for Sports Sciences of the University of Coimbra (CE/FCDEF-UC/00182016). Participants were recruited from clubs in Portugal. Informed consent was signed by parents or legal guardians and, subsequently, swimmers were informed about the objectives, methods, benefits, and risks of the study and

could withdraw from the experiment at any time. Data were collected following standardized conditions using the same equipment and by a single experienced investigator who accumulated >500 assessments.

## Sample

The calculation of sample size for hierarchical models considered two levels of variance (level 1: repetitions; level 2: participants). According to the literature [12, 13], estimated parameters and respective standard errors may be biased when the sample includes less than 30 cases and the ICC ranges 0.10–0.30 at level 2. The current study rests on a sample that includes 38 male adolescent swimmers aged 10.1–13.3 years. To determine intra-individual mean differences between the best PT extracted from each independent testing sessions, a subsample of swimmers performed a second visit to the laboratory. The sample size for the test-retest analysis was calculated using G*Power version 3.1.9.7 [14]. Considering the analysis to be tested on the primary outcome, a repeated-measures within factors anticipating a "large" effect size (f = 0.5), with an $\alpha$ = 0.05, a statistical power of (1 –$\beta$) = 0.95, a correlation (r) equal to 0.50 and a violation of sphericity ($\varepsilon$) = 0.80, would require 16 participants. With 17 participants enrolled in this secondary analysis, the sample size was then satisfactory. A previous study examined intra-individual changes in isokinetic strength among 19 healthy children [15]. Inclusion criteria in the current study were defined as follows: registered in the Portuguese Swimming Federation for at least two years, not being injured in the lower limbs during the past season. Chronological age (CA) was obtained as the difference between the laboratory visit date and birth date. Descriptive statistics are summarized in Table 1.

## Anthropometry

A single investigator measured body size following standardized protocols [16]. Stature was measured to the nearest 0.1 cm using a portable stadiometer (Harpenden model 98.603, Holtain LTD, Crosswell, UK). Body mass was measured to the nearest 0.1 kg using a portable balance (SECA model 770, Hanover, MD, USA).

## Isokinetic dynamometry

The isokinetic strength of the KE and KF was assessed on the preferred lower limb using a Biodex System 3 dynamometer (Shirley, NY, USA) adopting an angular velocity of 60˚.s$^{-1}$. The protocol started with a 5-minute warm-up performed on a cycle ergometer (814E Monark, Varberg, Sweden) with a resistance corresponding to 2% of the body mass [17] and a cadence of 50–60 rpm. Afterwards, the participant seated on the isokinetic dynamometer, and the lever arm was adjusted to fit the lateral condyle of the knee. Straps were positioned in the trunk, proximal thigh, and slightly above the medial malleolus of the tibia. The range of motion was defined as 85 degrees between 5 and 90 degrees. The participant was asked to perform a

**Table 1. Descriptive statistics on chronological age, training experience, body size among male adolescent swimmers by testing session.**

| Variable | units | Testing session | |
|---|---|---|---|
| | | first visit (n = 38) | second visit (n = 17) |
| Chronological age | years | 12.03±0.89 | 12.1±0.9 |
| Training experience | years | 3.8±1.6 | 4.2±1.8 |
| Stature | Cm | 151.4±8.8 | 150.6±7.3 |
| Body mass | kg | 41.7±8.1 | 40.2±7.2 |

voluntary maximal knee extension corresponding to the angular position of 0 degree. Then, the initial five degrees of flexion were suppressed to allow the participant to exert at least 10% of the defined torque limit. The effect of gravity was corrected as recommended by the manufacturer using the Biodex Advantage software package (Biodex Medical Systems, Shirley, NY, USA). The preceding occurred after weighing the member relaxed at the angular position of 30° [18]. As previously published, participants were instructed to keep their upper arms crossed and hands on the opposite shoulder during the test [17–19]. The protocol started with three continuous reciprocal concentric contractions of KE and KF for familiarization. After a 60-second interval, five reciprocal maximal concentric (cc) contractions of KE and KF were completed [20, 21]. Then, participants rested for 60 seconds and the sequence was repeated for the reciprocal eccentric (ecc) contractions of KF and KE. For each set (KEcc/KFcc; KFecc/KEecc), the five isokinetic curves were visually inspected before using the software AcqKnowledge 4.1 (Biopac Systems Incorporated, Goleta, CA, USA) to confirm data quality. Torque values were valid if obtained within 95% of the settled angular velocity, that is, between $57°.s^{-1}$ and $63°.s^{-1}$. The preceding routine allows the extraction of peak torque (PT) values, expressed in N.m, and respective angular position. PT refers to the highest value obtained in the range of motion of an isokinetic curve [21].

## Analyses

Descriptive statistics were calculated for the total sample. Intra-individual (5-repetitions) and inter-individual variances in PT values were tested using multilevel regression models. The preceding was done separately for KE and KF according to the type of contractions, that is concentric and eccentric. Repetitions were the fixed factor at level 1 and participants interpreted as level 2 (between individuals). In parallel, the intra-class correlation coefficient (ICC) was determined to examine intra-individual changes within the training session. Subsequently, the frequencies of the best PT across the five repetitions of the protocol were calculated to interpret the distribution of the best PT. Based on a subsample of 17 swimmers who visited the laboratory on a different day, intra-individual mean differences were determined in addition to a technical error of measurement and respective coefficient of variation for the best PT from the 5-repetition protocol. The difference between the best PT from the two testing sessions is interpreted as error (e). Technical error of measurement (TEM) is expressed in the same unit of the variable under analysis and refers to the square root of the quotient between the sum of squared errors and two times the number of participants in the repeated measurements [22]. Calculation was obtained as follows: $(sum(e^2)/2n)^{1/2}$. Intra-individual mean differences were interpreted after calculating Cohen d-value [23] as follows [24]: d<0.20 (trivial), 0.20<d<0.59 (small), 0.60<d<1.19 (moderate), 1.20<d<1.99 (large), 2.00<d<3.90 (very large). Finally, the Bland-Altman procedure [25] was used to examine the relationship between intra-individual agreements and the average of the repeated measures. Analyses were completed using MLwiN v2.26 (Center for Multilevel Modelling, University of Bristol, Bristol, UK), SPSS version 27.0 (SPSS Inc., IBM Company, N.Y., USA) and Graphpad Prism (version 5.00 for Windows, GraphPad Software, San Diego California USA, www.graphpad.com). The significance level was kept at 5%.

## Results

Descriptive statistics for PT separately for each of the five repetitions derived from the isokinetic strength protocol are presented in Table 2. The gradient for PT values was KEecc > KEcc > KFecc > KFcc. Multilevel regression models suggest intra-individual and inter-individual variability as significant sources of variance. ICC values fluctuated between 0.582 (KFcc) and

**Table 2. Descriptive statistics on isokinetic strength outputs for the 5-repetition protocol designed to assess knee muscle groups on concentric and eccentric contractions at 600.s$^{-1}$, in addition to multilevel regression models to examine intra and inter-individual variances (middle), plus ICC (right) among male adolescent swimmers (n = 38).**

| Variable | rep | n* | mean±SD | Multilevel regression models | | | ICC |
|---|---|---|---|---|---|---|---|
| | | | | Constant | e$_{ij}$ | u$_j$ | |
| KEcc (N.m) | | | | 67.8 (3.5) | 76.6 (8.8) | 385.8 (92.1) | 0.834 |
| | 1 | 37 | 67.7 ± 22.4 | | | | |
| | 2 | 37 | 80.6 ± 20.2 | | | | |
| | 3 | 38 | 82.5 ± 22.3 | | | | |
| | 4 | 38 | 81.3 ± 22.4 | | | | |
| | 5 | 38 | 80.9 ± 20.0 | | | | |
| KEecc (N.m) | | | | 91.0 (4.8) | 226.3 (26.4) | 649.6 (159.9) | 0.742 |
| | 1 | 37 | 90.7 ± 29.1 | | | | |
| | 2 | 38 | 86.9 ± 29.7 | | | | |
| | 3 | 38 | 89.6 ± 33.4 | | | | |
| | 4 | 36 | 89.5 ± 27.8 | | | | |
| | 5 | 36 | 80.5 ± 28.9 | | | | |
| KFcc (N.m) | | | | 41.7 (2.2) | 76.9 (8.9) | 107.1 (28.2) | 0.582 |
| | 1 | 38 | 41.7 ± 14.3 | | | | |
| | 2 | 36 | 42.0 ± 11.5 | | | | |
| | 3 | 37 | 40.9 ± 11.3 | | | | |
| | 4 | 38 | 41.2 ± 12.3 | | | | |
| | 5 | 38 | 42.9 ± 18.2 | | | | |
| KFecc (N.m) | | | | 61.2 (2.6) | 68.3 (8.3) | 189.4 (47.0) | 0.749 |
| | 1 | 38 | 61.2 ± 16.7 | | | | |
| | 2 | 34 | 59.2 ± 17.9 | | | | |
| | 3 | 35 | 60.7 ± 17.9 | | | | |
| | 4 | 35 | 57.6 ± 13.6 | | | | |
| | 5 | 33 | 55.8 ± 14.1 | | | | |

Note

* For each repetition, when PT was not obtained within 95% of the angular velocity, data was not retained for analysis.

Abbreviations: PT (peak torque); SD (standard deviation); e$_{ij}$ (Level 1 intra-individual variance); u$_j$ (Level 2 inter-individual variance); ICC (intra-class correlation); KEcc (knee extensors concentric); KEecc (knee extensors eccentric); KFcc (knee flexors concentric); KFecc (knee flexors eccentric).

0.834 (KEcc). According to Table 3, although the best PT tended to be extracted within the first three repetitions (KEccc: 63.1% of the cases; KEeccc: 76.3%; KFcc: 60.5% KFecc: 81.6%), a substantial number of participants attained their best PT on subsequent repetitions. As illustrated in Fig 1, the fluctuation of two consecutive repetitions tended to be trivial or small except for KEcc (showed an effect size d = 0.61 between the first and second repetitions with moderate increment). For the other muscle actions, the effect size was small or trivial.

The current study added information about the test-retest reliability of isokinetic strength assessments obtained in two independent testing sessions. As presented in Table 4, the magnitude of intra-individual mean differences tended to be small or trivial. Technical error of measurement ranged 5.8 N.m to 10.3 N.m. The corresponding coefficient of variation ranged between 9.0% and 12.2%.

As illustrated in Fig 2, Bland-Altman plots showed that despite slight improvements in mean values from session 1 to session 2, the intra-individual differences between testing sessions corresponded to a negligible BIAS (KEcc: +3.0 N.m; KEecc: +3.2 N.m; KFcc: +1.8 N.m; KEcc: +2.2 N.m). Also, several participants registered a decrement in their highest PT between

**Table 3. Absolute frequencies of the highest PT derived from five repetitions separately by knee muscle group and type of contraction among male adolescent swimmers (n = 38).**

| Muscle | contraction | Repetition | best peak torque | |
|---|---|---|---|---|
| | | | $f$ | % |
| Knee extensors | concentric | 1 | 3 | 7.9% |
| | | 2 | 14 | 36.8% |
| | | 3 | 7 | 18.4% |
| | | 4 | 7 | 18.4% |
| | | 5 | 7 | 18.4% |
| Knee extensors | eccentric | 1 | 16 | 42.1% |
| | | 2 | 7 | 18.4% |
| | | 3 | 6 | 15.8% |
| | | 4 | 8 | 21.1% |
| | | 5 | 1 | 2.6% |
| Knee flexors | concentric | 1 | 11 | 28.9% |
| | | 2 | 6 | 15.8% |
| | | 3 | 6 | 15.8% |
| | | 4 | 6 | 15.8% |
| | | 5 | 9 | 23.7% |
| Knee flexors | eccentric | 1 | 12 | 31.6% |
| | | 2 | 12 | 31.6% |
| | | 3 | 7 | 18.4% |
| | | 4 | 4 | 10.5% |
| | | 5 | 3 | 7.9% |

PT (peak torque); $f$ (absolute frequencies)

testing sessions: KEcc (4 cases), KEecc (4 cases), KFcc (4 cases), KFecc (8 cases). Finally, as evidenced in Fig 3, body mass was not correlated to errors (differences between testing sessions) which means that test-retest reliability is identical independently of expressing PT values in N. m or per unit of body mass (N.m⁻kg⁻¹).

## Discussion

The current study examined the learning effect of a 5-repetition isokinetic strength protocol designed to assess reciprocal concentric contractions of KE and KF in addition to a similar set of eccentric contractions among male adolescent swimmers aged 10.1–13.3 years. A substantial number of participants obtained the best PT while performing the fourth and fifth repetitions, despite the possible induction of fatigue. By inference, it is adequate to recommend five repetitions for each knee muscle group and type of contraction. The ICC values obtained in the present work do not support PT as a stable characteristic across the five repetitions within the same testing session, which applies to both knee muscle groups and the two type of contractions. Consequently, the reduction of repetitions to less than five consecutive contractions in a single testing session would correspond to a loss of information in the assessment of maximum voluntary contraction. Meanwhile, between testing sessions, the best PT values retained for analysis on each day seemed to have a negligible bias independently of knee muscle, although 9%-12% of the inter-individual variance in the best PT was affected by the intra-individual variance between days. This result was consistent with previously published studies [9]. Repeated measurements of the best PT confirmed a test-retest reliability. Thus, a practice day

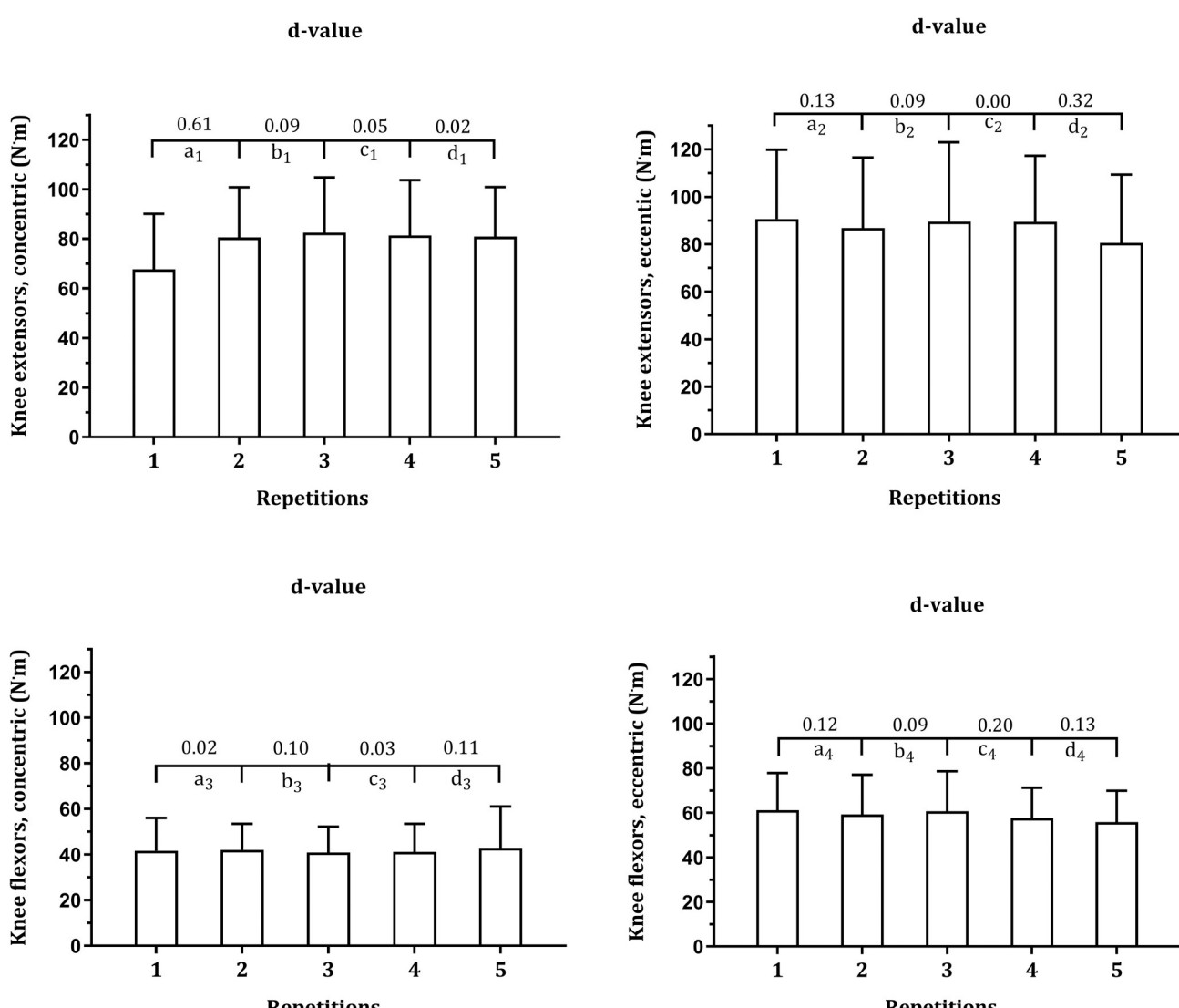

**Fig 1. Magnitude of mean differences on peak torque values by repetitions separately for muscle group and contraction among male adolescent swimmers (n = 38).**

**Table 4. Descriptive statistics for time-moments 1–2, intra-individual mean differences between time-moments, technical error of measurement, and magnitude effect between repeated measurements among male adolescent swimmers (n = 17).**

| | | Mean ± standard deviation | | Intra-individual differences | | TEM | %CV | Magnitude effect | |
|---|---|---|---|---|---|---|---|---|---|
| | | TM1 | TM2 | mean | (95% CL) | | | $d$ | (qualitative) |
| KEcc | N.m | 81.9±15.8 | 90.2±17.5 | 8.3 | (2.0;14.6) | 10.3 | 11.9 | 0.51 | (small) |
| KEecc | N.m | 101.8±22.2 | 110.2±19.3 | 8.3 | (1.5;15.2) | 10.9 | 10.3 | 0.42 | (small) |
| KFcc | N.m | 45.9±9.8 | 49.7±11.7 | 3.8 | (-0.0;7.7) | 5.8 | 12.2 | 0.36 | (small) |
| KFecc | N.m | 68.1±12.6 | 69.4±11.1 | 1.3 | (-3.3;5.9) | 6.2 | 9.0 | 0.11 | (trivial) |

KEcc (knee extensors concentric); KEecc (knee extensors eccentric); KFcc (knee flexors concentric); KFecc (knee flexors eccentric); TM1 (time-moment 1); TM2 (time-moment 2); TEM (technical error of measurement); CV (coefficient of variation); d (Cohen d-value).

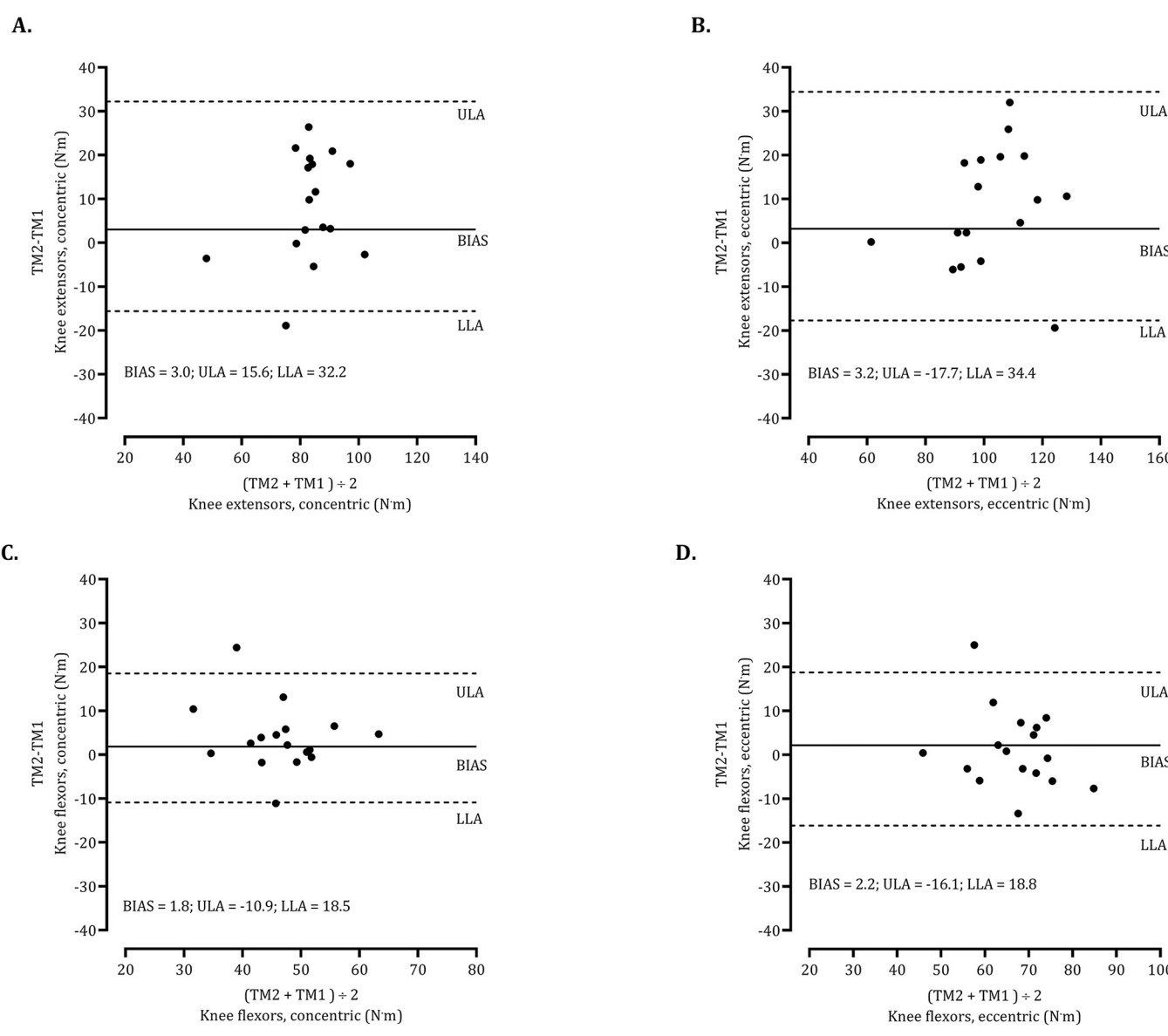

**Fig 2. Agreement between repeated measures for peak torque by muscle group and contraction.** Mean differences between time moments (TM2—TM1) and corresponding mean value [95% confidence intervals (95% CI)]. Dashed lines represent 95% limits of agreement (±1.96 SD); Bias are also represented as lower limits of agreement (LLA), and upper limits of agreement (ULA).

preceding evaluation of isokinetic knee strength may not be necessary when testing young swimmers.

The relationships between isokinetic strength assessment of knee joint muscle actions, somatic maturation, and body size were examined in male basketball players aged 14–16 years [26] who visited the laboratory for three occasions. The cited study reported an improvement of %CV derived from the second and a third testing sessions (4%-6%) compared to corresponding values derived from the two earliest visits (%CV: 8%-17%). Note, however, that the sample size of basketball players in the mentioned study was somewhat limited (n = 8). It was also concluded that the predicted time from age at peak height velocity contributed to the inter-individual variance in isokinetic strength outputs which means that %CV may also be a

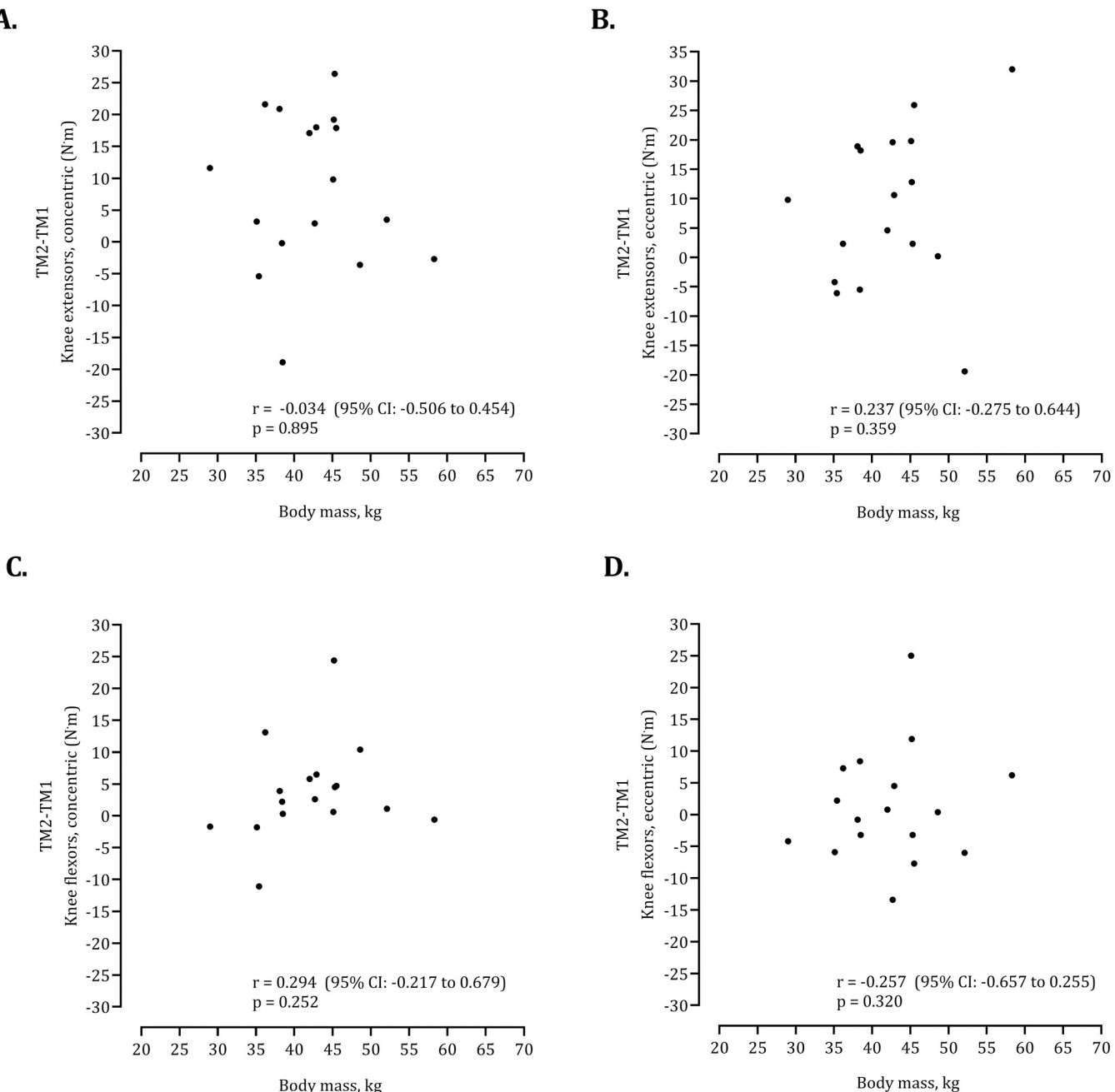

**Fig 3. Scatterplot of errors (differences between two testing sessions) and body mass including bivariate correlation coefficients.**

consequent of the sample homogeneity. The literature consistently indicates that the maximal force generated by skeletal muscles is primarily a function of muscle size [27].

The intra-individual fluctuation in isokinetic strength may be hypothesized as more problematic in youth than in trained adults [15] as recently demonstrated in a study of 32 trained male adults aged 27.9 years who were assessed on three occasions within 8 days to determine the variability of the PT of KE (using a protocol of five repetitions at a similar angular velocity

as in the present study). The authors observed a modest intra-individual variability of the highest PT between sessions (%CV ranged 6.4% to 15.3%). Previously [19], 26 male athletes aged 24.0±0.7 years from different sports completed two laboratory visits within a single week to determine the intra-individual variation of knee muscle actions assessed by isokinetic strength at an angular velocity of $60°·s^{-1}$. The coefficient of variation was lower than 5.2%, which was less than the values obtained in the current study who were exposed to the same protocol.

A recent study modelled longitudinal changes over a 5-year period in the isokinetic strength of the KE and the KF among 67 Belgian male adolescent (aged 11.0–13.9 years) soccer players [5]. According to their results, the isokinetic strength of the knee muscle groups may be reasonably predicted from chronological age, stature, and estimated fat-free mass with age *per se* contributing to an annual increment of 5.4 N.m in KE and 2.7 N.m in KF [5]. By obvious reasons, longitudinal studies should report data quality every time moment. Research in youth sports needs to integrate familiarization as a source of variation in addition to developmental changes. The literature [2] suggested KE as more reliable compared to KF, and concentric contractions as more likely to be reliable than eccentric contractions. In the current study, the gradient was identical on repeated occasions: KEecc > KEcc > KFecc > KFcc.

Meantime, the reliability of the isokinetic dynamometer in concentric passive mode was evaluated among 21 children (10 girls, 11 boys) aged 5–12 years who were tested twice within eight days [28]. They performed five consecutive repetitions of KE and KF at $60°.s^{-1}$ adopting the passive mode and uniquely the concentric contraction. The ICC values were 0.84 and 0.81 for KE and KF respectively. The preceding parameters were slightly higher than the respective coefficients calculated in the present study. Future research in youth sports needs to combine repeated measurements of passive and reactive modes.

The current study has limitations that need to be considered when interpreting the results. The protocol only considered the angular velocity of $60°.s^{-1}$. It may be hypothesized that familiarization would be distinct adopting other angular velocities. Additionally, the eccentric contraction was assessed following a reactive mode that is supposed to be different from the passive mode regarding the learning effect. The reactive protocol in the current study implied a reduction of the range of motion to avoid the initial degrees in the range of motion (zero degrees corresponds to full extension). Finally, future research may consider a larger sample for the two testing sessions. Probably the number of repetitions should be continued until further improvements would not be observed. Future studies may also test familiarization on dominant and non-dominant legs. Finally, although the present study did not support an association of intra-individual differences and body mass (see Fig 3), error may be standardized per unit of fat-free mass or lower-limb muscle mass to examine the potential relationship among error, morphological characteristics, growth, maturation and training experience.

## Conclusions

The number of repetitions to determine the best PT of KE and KF fluctuates between studies that also varied regarding the warm-up. According to the present study, assessing KF seems to be less stable across repetitions compared to KE. To obtain the best PT, it is not recommended to select protocols adopting less than 5 repetitions since a substantial number of participants exhibited their maximal performance in the last two repetitions.

## Supporting information

**S1 Data. .**
(XLSX)

## Author Contributions

**Conceptualization:** Daniela C. Costa, João Valente-dos-Santos, Rui Soles-Gonçalves, Manuel J. Coelho-e-Silva.

**Data curation:** Daniela C. Costa, João P. Duarte, Rui Soles-Gonçalves, Manuel J. Coelho-e-Silva.

**Formal analysis:** Daniela C. Costa, João Valente-dos-Santos, Diogo V. Martinho, Manuel J. Coelho-e-Silva.

**Investigation:** Daniela C. Costa, Paulo Sousa-e-Silva, João P. Duarte, Tomás Oliveira, Oscar M. Tavares, Joaquim M. Castanheira, Rui Soles-Gonçalves, Telmo Pereira, Jorge Conde, Daniel Courteix, Manuel J. Coelho-e-Silva.

**Methodology:** Daniela C. Costa, João Valente-dos-Santos, Jorge M. Celis-Moreno, Paulo Sousa-e-Silva, Diogo V. Martinho, Oscar M. Tavares, Joaquim M. Castanheira, Rui Soles-Gonçalves, Daniel Courteix, Manuel J. Coelho-e-Silva.

**Project administration:** Telmo Pereira, Jorge Conde, Daniel Courteix, Manuel J. Coelho-e-Silva.

**Resources:** Oscar M. Tavares, Joaquim M. Castanheira, Telmo Pereira, Jorge Conde, Daniel Courteix, Manuel J. Coelho-e-Silva.

**Software:** Rui Soles-Gonçalves, Manuel J. Coelho-e-Silva.

**Supervision:** Daniel Courteix, Manuel J. Coelho-e-Silva.

**Validation:** Daniela C. Costa, João Valente-dos-Santos, Jorge M. Celis-Moreno, Paulo Sousa-e-Silva, Diogo V. Martinho, João P. Duarte, Tomás Oliveira, Oscar M. Tavares, Joaquim M. Castanheira, Rui Soles-Gonçalves, Telmo Pereira, Jorge Conde, Daniel Courteix, David Thivel, Manuel J. Coelho-e-Silva.

**Visualization:** Paulo Sousa-e-Silva, Diogo V. Martinho, João P. Duarte, Tomás Oliveira, Oscar M. Tavares, Joaquim M. Castanheira, Rui Soles-Gonçalves, Telmo Pereira, Jorge Conde, Daniel Courteix, Manuel J. Coelho-e-Silva.

**Writing – original draft:** Daniela C. Costa, Diogo V. Martinho, Manuel J. Coelho-e-Silva.

**Writing – review & editing:** Daniela C. Costa, João Valente-dos-Santos, Jorge M. Celis-Moreno, Tomás Oliveira, Rui Soles-Gonçalves, Daniel Courteix, David Thivel, Manuel J. Coelho-e-Silva.

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
