## [Decision Letter · Decision Letter 0]

19 Dec 2022

PONE-D-22-24405Learning effect on an isokinetic knee strength test protocol among male adolescent athletesPLOS ONE

Dear Dr. Manuel J. Coelho-e Silva,

Thank you for submitting your manuscript to PLOS ONE. After careful consideration, we feel that it has merit but does not fully meet PLOS ONE’s publication criteria as it currently stands. Therefore, we invite you to submit a revised version of the manuscript that addresses the points raised during the review process.

Dear authors, I am sorry for the delay in making a decision regarding your article, I had a hard time finding appropriate reviewers. Both of the reviewers agreed that you need to introduce major changes prior to considering this article for publication. Please read carefully the comments provided below.All the best, Danica Janicijevic, academic editor

We look forward to receiving your revised manuscript.

Kind regards,

Danica Janicijevic, Ph.D

Academic Editor

PLOS ONE

https://journals.plos.org/plosone/s/fileid=ba62/PLOSOne_formatting_sample_title_authors_affiliations.pdf.

Reviewers' comments:

Reviewer's Responses to Questions

**Comments to the Author**

1. Is the manuscript technically sound, and do the data support the conclusions?

Reviewer #1: Yes

Reviewer #2: Partly

2. Has the statistical analysis been performed appropriately and rigorously? 

Reviewer #1: I Don't Know

Reviewer #2: Yes

3. Have the authors made all data underlying the findings in their manuscript fully available?

Reviewer #1: Yes

Reviewer #2: Yes

4. Is the manuscript presented in an intelligible fashion and written in standard English?

Reviewer #1: No

Reviewer #2: No

5. Review Comments to the Author

Reviewer #1: Congratulations on a well organized study.

Overall, I found the manuscript very interesting, and the study adds new information to this field.

However, I would recommend obtaining help from an English-speaking colleague to review the text and make changes throughout (i.e. meanwhile instead of meantime)

More specifically, here are further directed comments:

Introduction: did not convey "why" it is important to test strength in this age group, or why this study is needed right now.

lines 79-86: can combine as a range of examples in the literature, vs citing each individually. It takes too long to get to the point.

lines 94-95: need more evidence to substanciate this affirmation

Methods:

line 113: describe the training of the observer (have they done this many times?)

line 116: reconsider the format to express age, seems odd with two decimal points currently

lines 143-144: not familiar if this is the standard method to correct for gravity with this device. please add reference.

line 156: describe what would be considered an "atypical curve". Were there data removed in the final analysis?

lines 160-161: is the sample sufficient to power a multi-level regression analysis? If not, reconsider inclusion.

lines 168-171: if it was the intention to do a test-retest measurement, this should appear as a secondary goal at the end of the introduction

line 171: "d-value"? Does this mean Cohen's d value? please specify

Results:

line 189: wording needs to be softer, i.e. regression results "suggest" these are important factors

Table 3: add missing abbreviation explanations in the table notes

line 255: please reword so that you state that this is a hypothesis or a speculation

Discussion:

lines 273-274: please reword. Lacks clarity.

line 300-301: the comparison with your findings should appear much earlier in this paragraph. Needs rewriting.

lines 311-312: please include description of this subsample -> do they represent the original sample well?

General:

Would you consider it relevant to compare relative strength results standardized by weight as well? The differences between repetitions may become more trivial statistically, but your main findings showing that an important group of participants only exhibit their maximal capacities in the last repetitions would remain true.

Reviewer #2: PONE-D-22-24405

Learning effect on an isokinetic knee strength test protocol among male adolescent athletes

Prof. Manuel J. Coelho-e-Silva

PLOS ONE

COMMENT

The subject and results of presented study are of practical impotency for standardization of isokinetic dynamometer testing procedures. Invested effort for conducting and writing the paper is noticeable. However, some methodological corrections must be applied, as well as language improvements that are following almost the whole text, that should be revised and changed. I will address some of them with the following comments.

It is important to notice that results are not fully presented in conjunction with tables and figures. The authors should include more information and use more clear language to present obtained results.

COMMENT

Page 4, lines (97 – 101): From methodologic point of view the hypotheses are missing, which is leading to later methodological problems.

COMMENT

Page 4, line 108: Specify participants characteristics in this section. Age of participance should be standardized through whole paper, I suggest as mean and standard deviation.

COMMENT

Page 5, line 115: Sample should go before procedures?

COMMENT

Page 5, line 125: The name of reference is missing in a text form?

COMMENT

Page 5, line 131: This should be subchapter Experimental procedures?

COMMENT

Page 7, line 168: Subsample of 17 swimmers is not mentioned in sample subchapter, what are the specific characteristic of this sample, are there differences between this subgroup and mane sample?

Also in this case you have not precisely formulated the aim and hypothesize for determination of reliability of executed isokinetic test…

COMMENT

Page 7, line 171: Please specify what effect size measure have you used, Cohen’s d or some other?

COMMENT

Page 7, line 173: The Bland-Altman plot analysis is evaluating a bias between the mean differences, and to in your case estimate an agreement interval between repeated measurements. Suggestion is to correct differences into agreements.

COMMENT

Page 8, line (185): Table 1 should be separated into two tables, one with Descriptive morphological data (on chronological age, training experience, body size), and second one with isokinetic strength outputs. Also, Table 1 should have characteristics of subsample, or it can be in separated table.

COMMENT

Page 8, line 183: No need to repeat subjects characteristics.

COMMENT

Page 8, lines (220-228): It is very difficult the follow the description of results in correlation with figures and tables.

Lines 220 – 224: The text should in subchapter that describes sample and procedures.

Lines 227 – 228: What exactly several participants mean, how many of them exactly. This is not mentioned in discussion?

COMMENT

Page 11, lines (243-260): The authors should clarify the following section in correlation with presented hypotheses, aims, and results, to avoid confusion. The language should be revised to improve the readability.

Line 246: This is not in correlation with previous mentioned sample age.

COMMENT

Page 13, lines (299-300): There is no need to repeat the protocol.

COMMENT

Pages 3-12: Reference should be added to the end of sentences in next lines: 74, 75, 78, 92, 94, 96, 274.

COMMENT

Pages 3-12: Better sentence formulation: lines 75 – 78, 86 – 90, 95 – 96, 97 – 101, 109 – 112, 161 – 164.

6. PLOS authors have the option to publish the peer review history of their article (what does this mean?). If published, this will include your full peer review and any attached files.

Reviewer #1: No

Reviewer #2: No

---

## [Author Response · Author response to Decision Letter 0]

10 Mar 2023

Reviewers' comments:

Reviewer's Responses to Questions

Comments to the Author

1. Is the manuscript technically sound, and do the data support the conclusions?

Reviewer #1: Yes

Reviewer #2: Partly

2. Has the statistical analysis been performed appropriately and rigorously? 

Reviewer #1: I Don't Know

Reviewer #2: Yes

3. Have the authors made all data underlying the findings in their manuscript fully available?

Reviewer #1: Yes

Reviewer #2: Yes

4. Is the manuscript presented in an intelligible fashion and written in standard English?

Reviewer #1: No

Reviewer #2: No

5. Review Comments to the Author

REVIEWER #1: 

Congratulations on a well organized study. Overall, I found the manuscript very interesting, and the study adds new information to this field. However, I would recommend obtaining help from an English-speaking colleague to review the text and make changes throughout (i.e. meanwhile instead of meantime) More specifically, here are further directed comments:

AUTHORS: 

The revised version was edited by English-speaking colleague. Consequently, one author was added.

REVIEWER #1: 

Introduction: did not convey "why" it is important to test strength in this age group, or why this study is needed right now.

AUTHORS: 

The revised version now emphasizes the importance of data quality particularly in longitudinal studies and is citing two studies of young athletes adopting different number of repetitions to retain the best torque for analysis.

REVIEWER #1: 

lines 79-86: can combine as a range of examples in the literature, vs citing each individually. It takes too long to get to the point.

AUTHORS: 

The revised version was abbreviated and does not include information about warm-up and details of the sample in the cited studies.

REVIEWER #1: 

lines 94-95: need more evidence to substanciate this affirmation

AUTHORS: 

The revised version now includes an emphasis on the need to obtain reliable measurements in the context of youth sports.

Methods:

REVIEWER #1: 

line 113: describe the training of the observer (have they done this many times?)

Information added. The tests were performed by a single trained observer who accumulated more than 500 assessments.

AUTHORS: 

The information about the observer experience is now detailed.

REVIEWER #1: 

line 116: reconsider the format to express age, seems odd with two decimal points currently

AUTHORS: 

Adjusted

REVIEWER #1: 

lines 143-144: not familiar if this is the standard method to correct for gravity with this device. please add reference.

AUTHORS: 

Reference added.

REVIEWER #1: 

line 156: describe what would be considered an "atypical curve". Were there data removed in the final analysis?

Information added

AUTHORS: 

The sentence was adjusted, thanks.

REVIEWER #1:

lines 160-161: is the sample sufficient to power a multi-level regression analysis? If not, reconsider inclusion.

AUTHORS: 

Sample size included is justified in the respective section. Level has 190 measurements, level 2 has 38 cases.

REVIEWER #1: 

lines 168-171: if it was the intention to do a test-retest measurement, this should appear as a secondary goal at the end of the introduction

AUTHORS: 

The aim was adjusted following your suggestion, thanks

REVIEWER #1: 

line 171: "d-value"? Does this mean Cohen's d value? please specify

AUTHORS: 

Added

Results:

REVIEWER #1: 

line 189: wording needs to be softer, i.e. regression results "suggest" these are important factors

AUTHORS: 

Authors agree. Corrected

REVIEWER #1: 

Table 3: add missing abbreviation explanations in the table notes

AUTHORS: 

Missing abbreviations were introduced

REVIEWER #1: 

line 255: please reword so that you state that this is a hypothesis or a speculation

AUTHORS: 

Changed. Hopefully, it is now more understandable

Discussion:

REVIEWER #1: 

lines 273-274: please reword. Lacks clarity.

AUTHORS: 

Clarified. Thanks. 

REVIEWER #1: 

line 300-301: the comparison with your findings should appear much earlier in this paragraph. Needs rewriting.

AUTHORS: 

Modified accordingly

REVIEWER #1: 

lines 311-312: please include description of this subsample -> do they represent the original sample well?

AUTHORS: 

Mean values for chronological age, training experience, stature and body mass were now presented. They approached mean values of the total sample presented in Table 1

REVIEWER #1: 

General:

Would you consider it relevant to compare relative strength results standardized by weight as well? The differences between repetitions may become more trivial statistically, but your main findings showing that an important group of participants only exhibit their maximal capacities in the last repetitions would remain true.

AUTHORS: 

The conclusions were refined. The interesting note addressed by reviewer 1 would probably requires an allometric approach to the problem.

Reviewer #2: 

COMMENT

The subject and results of presented study are of practical impotency for standardization of isokinetic dynamometer testing procedures. Invested effort for conducting and writing the paper is noticeable. However, some methodological corrections must be applied, as well as language improvements that are following almost the whole text, that should be revised and changed. I will address some of them with the following comments. It is important to notice that results are not fully presented in conjunction with tables and figures. The authors should include more information and use more clear language to present obtained results.

AUTHORS: 

The manuscript in its revised version was improved according to comments addressed by both reviewers and English was revised by David Thivel.

Reviewer #2: 

COMMENT

Page 4, lines (97 – 101): From methodologic point of view the hypotheses are missing, which is leading to later methodological problems.

AUTHORS: 

Hypotheses were added at the end of the section introduction.

Reviewer #2: 

COMMENT

Page 4, line 108: Specify participants characteristics in this section. Age of participance should be standardized through whole paper, I suggest as mean and standard deviation.

AUTHORS: 

The range, mean and standard deviation was now included in the revised version.

Reviewer #2: 

COMMENT

Page 5, line 115: Sample should go before procedures?

AUTHORS: 

Thanks for the question. Authors do not agree and in all past manuscripts (including in BMC Pediatrics) the organization of the experiment including approval in ethics committee, distribution of informed consent precedes the sample. Let us wait for the editor-in-chief recommendations. If needed, authors will modify the structure of the methods section 

Reviewer #2: 

COMMENT

Page 5, line 125: The name of reference is missing in a text form?

AUTHORS: 

Corrected. Hopefully, it is now understandable for readers.

Reviewer #2: 

COMMENT

Page 5, line 131: This should be subchapter Experimental procedures?

Anthropometry and isokinetic dynamometer are present now as subchapters of procedures.

AUTHORS: 

Let us wait for the editor-in-chief recommendations. If needed, authors will modify the structure of the methods section

Reviewer #2: 

COMMENT

Page 7, line 168: Subsample of 17 swimmers is not mentioned in sample subchapter, what are the specific characteristic of this sample, are there differences between this subgroup and mane sample?

AUTHORS: 

Now added. Thanks.

Reviewer #2: 

Also in this case you have not precisely formulated the aim and hypothesize for determination of reliability of executed isokinetic test…

AUTHORS: 

The final sentence of the last paragraph in the introduction was revised accordingly to this comment. Thanks.

Reviewer #2: 

COMMENT

Page 7, line 171: Please specify what effect size measure have you used, Cohen’s d or some other?

It was added.

AUTHORS: 

Reviewer #2: 

COMMENT

Page 7, line 173: The Bland-Altman plot analysis is evaluating a bias between the mean differences, and to in your case estimate an agreement interval between repeated measurements. Suggestion is to correct differences into agreements.

AUTHORS: 

The term “differences” was replaced by “agreements” as suggested.

Reviewer #2: 

COMMENT

Page 8, line (185): Table 1 should be separated into two tables, one with Descriptive morphological data (on chronological age, training experience, body size), and second one with isokinetic strength outputs. Also, Table 1 should have characteristics of subsample, or it can be in separated table.

AUTHORS: 

Modified as suggested.

Reviewer #2: 

COMMENT

Page 8, line 183: No need to repeat subjects characteristics.

AUTHORS: 

Characteristics of the participants were now concentrated in Table 1 and are not repeated in the text.

Reviewer #2: 

COMMENT

Page 8, lines (220-228): It is very difficult the follow the description of results in correlation with figures and tables.

AUTHORS: 

Hopefully the revised version is now easier to follow.

Reviewer #2: 

Lines 220 – 224: The text should in subchapter that describes sample and procedures.

AUTHORS: 

Done

Reviewer #2: 

Lines 227 – 228: What exactly several participants mean, how many of them exactly. This is not mentioned in discussion?

AUTHORS: 

Detailed information about the number of cases who decreased their highest PT was introduced.

Reviewer #2: 

 Page 11, lines (243-260): The authors should clarify the following section in correlation with presented hypotheses, aims, and results, to avoid confusion. The language should be revised to improve the readability.

AUTHORS: 

Improved

Reviewer #2: 

Line 246: This is not in correlation with previous mentioned sample age.

AUTHORS: 

Corrected

Reviewer #2: 

COMMENT

Page 13, lines (299-300): There is no need to repeat the protocol.

AUTHORS: 

Thanks. Repetition of the protocol was deleted. 

Reviewer #2: 

COMMENT

Pages 3-12: Reference should be added to the end of sentences in next lines: 74, 75, 78, 92, 94, 96, 274.

AUTHORS: 

Added accordingly to recommendations

Reviewer #2: 

COMMENT

Pages 3-12: Better sentence formulation: lines 75 – 78, 86 – 90, 95 – 96, 97 – 101, 109 – 112, 161 – 164.

AUTHORS: 

Refined by Professor David Thivel who was introduced as co-author. Hopefully, the manuscript is now understandable.

---

## [Decision Letter · Decision Letter 1]

15 May 2023

PONE-D-22-24405R1Learning effect on an isokinetic knee strength test protocol among male adolescent athletesPLOS ONE

Dear Dr. Coelho-e-Silva,

Thank you for submitting your manuscript to PLOS ONE. After careful consideration, we feel that it has merit but does not fully meet PLOS ONE’s publication criteria as it currently stands. Therefore, we invite you to submit a revised version of the manuscript that addresses the points raised during the review process.

ACADEMIC EDITOR: Dear Authors, two experts in the field reviewed your revised manuscript version founding some minor points you should consider in the revision process.

We look forward to receiving your revised manuscript.

Kind regards,

Emiliano Cè

Academic Editor

PLOS ONE

Journal Requirements:

Additional Editor Comments (if provided):

Reviewers' comments:

Reviewer's Responses to Questions

**Comments to the Author**

1. If the authors have adequately addressed your comments raised in a previous round of review and you feel that this manuscript is now acceptable for publication, you may indicate that here to bypass the “Comments to the Author” section, enter your conflict of interest statement in the “Confidential to Editor” section, and submit your "Accept" recommendation.

Reviewer #1: (No Response)

Reviewer #2: All comments have been addressed

2. Is the manuscript technically sound, and do the data support the conclusions?

Reviewer #1: Partly

Reviewer #2: Yes

3. Has the statistical analysis been performed appropriately and rigorously? 

Reviewer #1: Yes

Reviewer #2: Yes

4. Have the authors made all data underlying the findings in their manuscript fully available?

Reviewer #1: Yes

Reviewer #2: Yes

5. Is the manuscript presented in an intelligible fashion and written in standard English?

Reviewer #1: No

Reviewer #2: Yes

6. Review Comments to the Author

Reviewer #1: Thank you for editing this new version of your manuscript with careful consideration for the comments from both reviewers. This version is a very significant improvement for international readers.

I would recommend further attention to the following comments for the next round:

In general, try to decrease the length of your sentences to avoid syntax errors and to make reading easier to follow for your audience.

- Objective 2: state as test-retest reliability for clarity. Current wording makes it harder to distinguish from first objective.

- line 78: "meantime" should be "meanwhile" (as per round 1 comments)

- although improved, paragraph 1 of the introduction still for me does not convey the importance of your main point: protocols differ between studies, sometimes 3 or 5 (or more). Validity of shorter protocols may be compromised if the peak torque is not obtained in the first three attempts.

- lines 95 to 103: needs rewriting. The content is there, but syntax is not at the correct level

- line 111: avoid using reference [15] about multiple sclerosis. This population has specific fatigue problems related to their neurological condition that cannot be compared with healthy youth swimmers

- lines 119/120: sentence appears incomplete. add what the ICC values are?

- lines 130/131: as previously mentioned, use term test-retest reliability for the international audience

- lines 155/159: the sample size calculation here does not work. You are using the number of measurements instead of the number of participants. The sentence described is coherent to evaluate the sample needed to power the test-retest comparison. You also need to state what the expected difference (effect size) would be between groups to justify the sample size.

- lines 159-164: good information, but should be moved to either the introduction or the discussion

- line 183: use the term "height" instead of stature for international audience. Change in Table 1 as well

- line 196: "strips" should be the word "straps"

- line 210: how was the data filtered?

- lines 214-215: this seems backwards. Would normally expect the raw data to be filtered to remove noise, then corrected for limb inertia, and then select only the data from the target speed (+/- 5%). Described as such, it seems you followed a different process.

- line 229: define what you mean by "technical error of measurement"

- lines 250-252: leave interpretation sentences for the discussion, does not belong in the Results

- Table 1: much improved, good to see the information for both groups here

- Table 2: must explain why there is not consistently the same number of participants for each rep? Is this related to the data processing question from lines 214-215? How do you explain that the ICC for KFcc is lower despite the values being more similar across the 5 repetitions?

- lines 301/309: paragraph is not clear enough, needs to be rewritten

- lines 311/313: link this information with your findings. it seems lost otherwise

- line 339: i believe you are presenting your values, please make it explicit for the reader

- lines 372/374: I appreciate that you considered my comment about relative strength measurements, but I would like you to confirm that you have proceeded with this calculation and confirmed that your statistical conclusions remain the same

Reviewer #2: COMMENT

Thank the authors for taking previous review advisees in preparing the presented study for publishing. Significant changes and improvements are noticeable. I will only suggest a few more corrections. The most important advice is to explain more about the differences between obtained results with the previous studies regarding both experimental questions. Father, to explain in more detail why the obtained mean difference and %CV between the first and second testing in KEcc and KEecc, is higher (is there a significant difference beside small ES)?

Overall, the study has provided exciting and valuable data regarding standardizing the protocol of isokinetic testing of youth athletes.

Some additional corrections and suggestions:

COMMENTS

Page 5, line 120: Can you please specify the range of ICC in previous studies, and type of motoric tasks or activity?

Page 6, line 159: There is no need to repeat the number of subjects.

Page 6-7, lines 159 - 164: This part should be in introduction. The number of the reference can be added to justify the sample size power.

Page 7, line 172: Can you please define how many days past between testing days (mean)?

Page 7, table 1: Suggestion - present as “mean±SD” instead “Mean Standard deviation”

Page 8, line 183: Without “described elsewhere” in sentence.

Page 8, line 184-185: A portable scale was used for measuring of body mass, please make the sentence clearer?

7. PLOS authors have the option to publish the peer review history of their article (what does this mean?). If published, this will include your full peer review and any attached files.

Reviewer #1: **Yes: **Félix Croteau

Reviewer #2: No

---

## [Author Response · Author response to Decision Letter 1]

14 Jun 2023

PONE-D-22-24405R1

Learning effect on an isokinetic knee strength test protocol among male adolescent athletes

Dear Dr. Coelho-e-Silva,

Thank you for submitting your manuscript to PLOS ONE. After careful consideration, we feel that it has merit but does not fully meet PLOS ONE’s publication criteria as it currently stands. Therefore, we invite you to submit a revised version of the manuscript that addresses the points raised during the review process.

ACADEMIC EDITOR: 

Dear Authors, 

two experts in the field reviewed your revised manuscript version founding some minor points you should consider in the revision process.

We look forward to receiving your revised manuscript.

Kind regards,

Emiliano Cè

Academic Editor

PLOS ONE

Journal Requirements:

AUTHORS:

The revised version was carefully checked.

Reviewer's Responses to Questions

Comments to the Author

1. If the authors have adequately addressed your comments raised in a previous round of review and you feel that this manuscript is now acceptable for publication, you may indicate that here to bypass the “Comments to the Author” section, enter your conflict of interest statement in the “Confidential to Editor” section, and submit your "Accept" recommendation.

Reviewer #1: (No Response)

Reviewer #2: All comments have been addressed

2. Is the manuscript technically sound, and do the data support the conclusions?

Reviewer #1: Partly

Reviewer #2: Yes

3. Has the statistical analysis been performed appropriately and rigorously?

Reviewer #1: Yes

Reviewer #2: Yes

4. Have the authors made all data underlying the findings in their manuscript fully available?

Reviewer #1: Yes

Reviewer #2: Yes

5. Is the manuscript presented in an intelligible fashion and written in standard English?

Reviewer #1: No

Reviewer #2: Yes

6. Review Comments to the Author

Reviewer #1: Thank you for editing this new version of your manuscript with careful consideration for the comments from both reviewers. This version is a very significant improvement for international readers.

AUTHORS:

Thanks.

I would recommend further attention to the following comments for the next round:

In general, try to decrease the length of your sentences to avoid syntax errors and to make reading easier to follow for your audience.

AUTHORS:

Authors have reduced the sentences and the manuscript was edited by David Thivel who obtained the PhD in Canada.

- Objective 2: state as test-retest reliability for clarity. Current wording makes it harder to distinguish from first objective.

AUTHORS:

Refined as suggested.

- line 78: "meantime" should be "meanwhile" (as per round 1 comments)

AUTHORS:

Changed as suggested.

- although improved, paragraph 1 of the introduction still for me does not convey the importance of your main point: protocols differ between studies, sometimes 3 or 5 (or more). Validity of shorter protocols may be compromised if the peak torque is not obtained in the first three attempts.

AUTHORS:

Thanks for the comment. The essential is now in the final sentence of the abstract and adequately commented across the manuscript. The revised version is now more objective and concise.

- lines 95 to 103: needs rewriting. The content is there, but syntax is not at the correct level

AUTHORS:

Adjusted as recommended.

- line 111: avoid using reference [15] about multiple sclerosis. This population has specific fatigue problems related to their neurological condition that cannot be compared with healthy youth swimmers

AUTHORS:

Deleted.

- lines 119/120: sentence appears incomplete. add what the ICC values are?

AUTHORS:

Adjusted as recommended.

- lines 130/131: as previously mentioned, use term test-retest reliability for the international audience

AUTHORS:

Changed as recommended across the entire manuscript.

- lines 155/159: the sample size calculation here does not work. You are using the number of measurements instead of the number of participants. The sentence described is coherent to evaluate the sample needed to power the test-retest comparison. You also need to state what the expected difference (effect size) would be between groups to justify the sample size.

AUTHORS:

Information regarding calculation of power sample is now included for the various analysis.

- lines 159-164: good information, but should be moved to either the introduction or the discussion

AUTHORS:

Done as suggested.

- line 183: use the term "height" instead of stature for international audience. Change in Table 1 as well

AUTHORS:

Stature refers to the specific height of the vertex. In all previous studies, authors adopt the term stature. 

- line 196: "strips" should be the word "straps"

AUTHORS:

Thanks. Changed.

- line 210: how was the data filtered?

- lines 214-215: this seems backwards. Would normally expect the raw data to be filtered to remove noise, then corrected for limb inertia, and then select only the data from the target speed (+/- 5%). Described as such, it seems you followed a different process.

AUTHORS: 

The revised version now includes the information.

- line 229: define what you mean by "technical error of measurement"

AUTHORS:

Detailed.

- lines 250-252: leave interpretation sentences for the discussion, does not belong in the Results

AUTHORS:

Agree. Deleted.

- Table 1: much improved, good to see the information for both groups here

AUTHORS:

Thanks.

- Table 2: must explain why there is not consistently the same number of participants for each rep? Is this related to the data processing question from lines 214-215? How do you explain that the ICC for KFcc is lower despite the values being more similar across the 5 repetitions?

AUTHORS:

The following was added in Table 2: * For each repetition, when PT was not obtained within 95% of the settled angular velocity, data was not retained for analysis.

Multilevel structures do not require balanced data to obtain eﬃcient estimates. In other words, it is not necessary to have the same number of lower-level units within each higher-level unit. With repeated measures data we do not require the same number of measurement occasions per individual subject (level 2).

In this type of model (i.e., variance components model), the residual variance is partitioned into components corresponding to each 2 level in the hierarchy and the variance between repetition and the variance between individuals within a given repetition are obtained. ICC, provides the proportion of the total residual variation.

Briefly, the apparently stable mean values for each repetitions combined to lower ICC for KFcc, indicate large intra-individual fluctuation of performance scores and inter-individual variance. The PT value is not fully stable across the five repetitions.

- lines 301/309: paragraph is not clear enough, needs to be rewritten

AUTHORS:

Improved, substantially.

- lines 311/313: link this information with your findings. it seems lost otherwise

AUTHORS:

linked.

- line 339: i believe you are presenting your values, please make it explicit for the reader

AUTHORS:

It is now explicit.

- lines 372/374: I appreciate that you considered my comment about relative strength measurements, but I would like you to confirm that you have proceeded with this calculation and confirmed that your statistical conclusions remain the same

AUTHORS:

Figures were revised and now refers to error in N.m per unit of body mass, that is, N.m/kg

Reviewer #2: COMMENT

Thank the authors for taking previous review advisees in preparing the presented study for publishing. Significant changes and improvements are noticeable. I will only suggest a few more corrections. The most important advice is to explain more about the differences between obtained results with the previous studies regarding both experimental questions. Father, to explain in more detail why the obtained mean difference and %CV between the first and second testing in KEcc and KEecc, is higher (is there a significant difference beside small ES)?

Overall, the study has provided exciting and valuable data regarding standardizing the protocol of isokinetic testing of youth athletes.

AUTHORS:

Thanks. In fact, Table 2 confirms that the magnitude of changes in mean values between testing sessions is negligible (small). This is well illustrated in Figures. However, %CV is larger for knee extensors than for knee flexors. Figure 2 has now identical Y-axis for KE and KF. This is now adequately commented in the main findings at discussion section.

Some additional corrections and suggestions:

COMMENTS

Page 5, line 120: Can you please specify the range of ICC in previous studies, and type of motoric tasks or activity?

AUTHORS:

The range of ICCs and type of activities were added to the introduction. Thanks.

Page 6, line 159: There is no need to repeat the number of subjects.

AUTHORS:

Adjusted.

Page 6-7, lines 159 - 164: This part should be in introduction. The number of the reference can be added to justify the sample size power.

AUTHORS:

The sentence was moved to the introduction. The following reference was added to justify the sample size used on the paired-t test analysis:

van Tittelboom, Vanessa et al. “Reliability of Isokinetic Strength Assessments of Knee and Hip Using the Biodex System 4 Dynamometer and Associations With Functional Strength in Healthy Children.” Frontiers in sports and active living vol. 4 817216. 24 Feb. 2022, doi:10.3389/fspor.2022.817216

Page 7, line 172: Can you please define how many days past between testing days (mean)?

AUTHORS:

This point is now clarified.

Page 7, table 1: Suggestion - present as “mean±SD” instead “Mean Standard deviation”

AUTHORS:

Adjusted.

Page 8, line 183: Without “described elsewhere” in sentence.

AUTHORS:

Adjusted.

Page 8, line 184-185: A portable scale was used for measuring of body mass, please make the sentence clearer?

AUTHORS:

The sentence was adjusted.

7. PLOS authors have the option to publish the peer review history of their article (what does this mean?). If published, this will include your full peer review and any attached files.

Do you want your identity to be public for this peer review? For information about this choice, including consent withdrawal, please see our Privacy Policy.

Reviewer #1: Yes: Félix Croteau

Reviewer #2: No

---

## [Decision Letter · Decision Letter 2]

26 Jun 2023

Learning effect on an isokinetic knee strength test protocol among male adolescent athletes

PONE-D-22-24405R2

Dear Dr. Coelho-e-Silva,

We’re pleased to inform you that your manuscript has been judged scientifically suitable for publication and will be formally accepted for publication once it meets all outstanding technical requirements.

Kind regards,

Emiliano Cè

Academic Editor

PLOS ONE

Additional Editor Comments (optional):

Reviewers' comments:

Reviewer's Responses to Questions

**Comments to the Author**

1. If the authors have adequately addressed your comments raised in a previous round of review and you feel that this manuscript is now acceptable for publication, you may indicate that here to bypass the “Comments to the Author” section, enter your conflict of interest statement in the “Confidential to Editor” section, and submit your "Accept" recommendation.

Reviewer #1: All comments have been addressed

Reviewer #2: All comments have been addressed

2. Is the manuscript technically sound, and do the data support the conclusions?

Reviewer #1: Yes

Reviewer #2: Yes

3. Has the statistical analysis been performed appropriately and rigorously? 

Reviewer #1: Yes

Reviewer #2: Yes

4. Have the authors made all data underlying the findings in their manuscript fully available?

Reviewer #1: Yes

Reviewer #2: Yes

5. Is the manuscript presented in an intelligible fashion and written in standard English?

Reviewer #1: Yes

Reviewer #2: Yes

6. Review Comments to the Author

Reviewer #1: I am fully satisfied with the modifications that you made on this final revision. This study will bring important knowledge to the field of isokinetic testing.

Minor adjustments for publishing:

line 93 in the introduction: change "probably" for "potentially". As per your results, many people still get their peak values in the early repetitions

line 363: change "By obvious reasons" for "therefore"

line 371: change "meantime" for "meanwhile"

line 388: remove "probably"

Reviewer #2: Thank the authors for taking previous review advisees in preparing the presented study for publishing. The study has provided exciting and valuable data regarding standardizing the protocol of isokinetic testing of youth athletes.

I wish a lot of success in a future research work.

7. PLOS authors have the option to publish the peer review history of their article (what does this mean?). If published, this will include your full peer review and any attached files.

Reviewer #1: No

Reviewer #2: No

---

## [Editor Report · Acceptance letter]

18 Jul 2023

PONE-D-22-24405R2 

Learning effect on an isokinetic knee strength test protocol among male adolescent athletes 

Dear Dr. Coelho-e-Silva:

I'm pleased to inform you that your manuscript has been deemed suitable for publication in PLOS ONE. Congratulations! Your manuscript is now with our production department. 

Kind regards, 

on behalf of

Prof. Emiliano Cè 

Academic Editor

PLOS ONE